# OpenReview forum: "CROW: Eliminating Backdoors from Large Language Models via Internal Consistency Regularization"
_ICML.cc/2025/Conference — ICML 2025 poster_

### Official Review · Reviewer_BKBt · 2025-03-10

**Overall Recommendation:** 3

**Summary:**

This paper introduces Internal Consistency Regularization (CROW), a defense mechanism against backdoor attacks in LLMs. Unlike traditional defenses, which are primarily designed for classification tasks, CROW effectively mitigates backdoors in text generation models without relying on clean reference models or prior knowledge of attack triggers. The core insight behind CROW is that backdoored LLMs exhibit inconsistent hidden representations across layers, whereas clean models maintain smooth and stable transitions. To counteract this instability, CROW enforces internal consistency by integrating adversarial perturbations and regularization during fine-tuning. Experimental results demonstrate that CROW preserves generative performance while significantly reducing the attack success rate (ASR) across five LLM architectures and six diverse backdoor attack strategies.

**Claims And Evidence:**

The experimental results support the claims presented in the submission. However, there appears to be a minor inconsistency between the authors' initial observation and the proposed methodology. Specifically, Figure 1 highlights that the primary discrepancy between backdoor-triggered and clean inputs manifests in the initial latent representations of the backdoored model, as indicated by the cosine similarity difference. This observation suggests that consistency regularization would be more naturally applied between the original latent representation and its perturbed counterpart, rather than across consecutive layers, as implemented in the proposed approach.

To strengthen the alignment between the method and the observed phenomenon, I recommend that the authors conduct experiments incorporating this alternative regularization strategy. If enforcing consistency across layers remains the preferred approach, it would be beneficial for the authors to provide additional empirical evidence demonstrating why this design choice is justified in light of their initial findings.

**Essential References Not Discussed:**

1. The use of adversarial perturbations for defending against backdoor attacks is not a novel approach, as several prior works [1,2,3] have successfully leveraged this property to mitigate backdoor vulnerabilities.

2. While this work proposes enforcing consistency among internal representations, prior research [4] has explored a similar direction by utilizing reference models to remove backdoor-related information.


References:
[1] Gao, Yinghua, et al. "On the effectiveness of adversarial training against backdoor attacks." IEEE Transactions on Neural Networks and Learning Systems (2023).

[2] Ali, Hassan, et al. "Adversarially Guided Stateful Defense Against Backdoor Attacks in Federated Deep Learning." arXiv preprint arXiv:2410.11205 (2024).

[3] Yang, Wenkai, et al. "RAP: Robustness-Aware Perturbations for Defending against Backdoor Attacks on NLP Models." Proceedings of the 2021 Conference on Empirical Methods in Natural Language Processing. 2021.

[4] Zhao, Shuai, et al. "Unlearning backdoor attacks for llms with weak-to-strong knowledge distillation." arXiv preprint arXiv:2410.14425 (2024).

**Experimental Designs Or Analyses:**

1. While CleanGen has certain constraints, it would be valuable to include a comparison, as it approaches backdoor defense from a different perspective. Such a comparison could provide a more comprehensive understanding of the strengths and limitations of each method.

2. The current VPI setting only considers the insertion-based scenario. However, the original paper also explores the use of entity-based triggers, which are more semantically coherent than abrupt insertions. Additionally, prior work [1] has demonstrated that semantic-level triggers tend to be more effective compared to insertion-based triggers. Incorporating this perspective would provide a more nuanced evaluation of defense robustness.

References:

[1] He, Xuanli, et al. "TuBA: Cross-Lingual Transferability of Backdoor Attacks in LLMs with Instruction Tuning." arXiv preprint arXiv:2404.19597 (2024).

**Methods And Evaluation Criteria:**

Yes

**Other Comments Or Suggestions:**

None

**Other Strengths And Weaknesses:**

Strengths:
The proposed method removes a strong assumption commonly made by previous defense techniques, such as the necessity of a clean reference model or prior knowledge of trigger patterns. This makes the approach more practical and broadly applicable across different backdoor attack scenarios.

**Questions For Authors:**

None

**Relation To Broader Scientific Literature:**

Please refer to **Essential References Not Discussed**

**Theoretical Claims:**

Yes

---

> ### Author Rebuttal · Authors · 2025-04-01
>
> # Response to Reviewer BKBt
>
> We sincerely appreciate your thorough review and insightful comments. Please find our responses to your questions below.
>
> **Q1.** Consistency regularization between original and perturbed embeddings
>
> **A1.** We appreciate your insightful suggestion. In response, we conducted additional experiments that apply consistency only between the original and perturbed embedding (i.e., the initial latent representation) and compare it to our layer-wise approach (CROW). Below is the expanded table:
>
> |Task|Attack|No Defense|Alternative|CROW|
> |-|-|:-:|:-:|:-:|
> |Sentiment|BadNets|65.00/2.72|2.56/4.13|0.53/3.80|
> |Sentiment|CTBA|63.33/2.80|9.29/4.17|2.08/3.80|
> |Refusal|BadNets|94.50/4.35|98.00/4.34|19.63/4.15|
> |Refusal|CTBA|82.16/4.40|30.22/4.42|2.38/4.27|
>
> *Note: ASR(↓)/MT-Bench(↑)*
>
> These results demonstrate that while initial embedding differences are significant, forcing consistency solely at this level cannot prevent residual perturbations from amplifying through deeper layers. Our layer-wise approach achieves consistent performance across all tasks while maintaining comparable utility. These findings confirm that comprehensive layer-wise constraints are essential for robust backdoor elimination.
>
> **Q2.** CleanGen
>
> **A2.** Following your suggestion, we provide comprehensive comparison results between our CROW and CleanGen on backdoored Llama-2-7B. As shown in the Table below, CleanGen achieves complete mitigation for Sentiment tasks while maintaining high MT-Bench scores. However, its effectiveness is less stable on the more challenging refusal-generation tasks. In contrast, CROW achieves more consistent mitigation across all task types, while maintaining comparable utility. We will include these experimental results in the revision.
>
> *Note: ASR(↓)/MT-Bench(↑)*
> |Task|Attack|No Defense|CleanGen|CROW|
> |-|-|:-:|:-:|:-:|
> |Sentiment|BadNets|65.00/2.72|0.00/4.81|0.53/3.80|
> |Sentiment|CTBA|63.33/2.80|0.00/4.87|2.08/3.80|
> |Sentiment|MTBA|18.56/1.00|0.00/4.81|0.00/3.89|
> |Sentiment|Sleeper|5.08/2.97|0.00/4.91|0.00/3.68|
> |Sentiment|VPI|13.79/3.08|0.00/4.86|0.00/3.69|
> |Refusal|BadNets|94.50/4.35|11.00/4.82|19.63/4.15|
> |Refusal|CTBA|82.16/4.40|53.50/4.86|2.38/4.27|
> |Refusal|MTBA|89.90/4.43|55.50/4.93|0.54/3.89|
> |Refusal|Sleeper|54.91/4.42|45.50/4.92|0.56/4.37|
> |Refusal|VPI|98.99/4.36|67.00/4.85|0.50/4.28|
>
> **Q3.** Semantic/entity-based VPI triggers
>
> **A3.** Thank you for raising this important point about entity-based triggers. We have addressed this concern in detail in our response to Reviewer 3BdD (Q3), where we present our evaluation against semantic backdoor attacks, including VPI-semantic [1] and semantic-level instruction backdoors [2]. As noted there, our experiments demonstrate CROW's effectiveness against these more semantically coherent triggers, with significant ASR reductions. This aligns with observations from recent literature [3] on the cross-lingual transferability of backdoor attacks, further supporting our approach's robustness against various trigger mechanisms.
>
> [1] Backdooring Instruction-Tuned Large Language Models with Virtual Prompt Injection. arXiv:2307.16888 2024.
>
> [2] Instruction Backdoor Attacks Against Customized LLMs. arXiv:2402.09179 2024.
>
> [3] TuBA: Cross-Lingual Transferability of Backdoor Attacks in LLMs with Instruction Tuning. arXiv:2404.19597 2024.
>
> **Q4.** Adversarial perturbations for backdoor defense [1,2,3], and similar internal representation consistency [4]
>
> **A4.** While prior works [1–3] leverage adversarial perturbations for backdoor defense in various settings, and [4] explores internal representation consistency via knowledge distillation, our method CROW differs from these approaches in several key aspects:
>
> (1) Target domain: CROW is designed for generative LLMs, whereas [1] focuses on vision models, [2] on federated learning, and [3] on classification-based NLP models.
>
> (2) Training vs. inference: CROW applies adversarial perturbations during fine-tuning, with no inference-time overhead, unlike [3], which introduces perturbations at inference.
>
> (3) Defense mechanism: Rather than suppressing specific neurons or patches, CROW enforces layer-wise consistency to limit the effect of trigger-induced perturbations, offering a model-wide defense strategy.
>
> (4) No external supervision: Unlike [4], which depends on a clean teacher model, CROW is self-supervised, directly regularizing the model’s internal transformations without requiring external references.
>
> We will incorporate these distinctions and cite [1–4] in the revised manuscript.
>
> [1] On the effectiveness of adversarial training against backdoor attacks. TNNLS 2023.
>
> [2] Adversarially Guided Stateful Defense Against Backdoor Attacks in Federated Deep Learning. arXiv:2410.11205 2024.
>
> [3] RAP: Robustness-Aware Perturbations for Defending against Backdoor Attacks on NLP Models. EMNLP 2021.
>
> [4] Unlearning backdoor attacks for llms with weak-to-strong knowledge distillation. arXiv:2410.14425 2024.

---

### Official Review · Reviewer_8Evs · 2025-03-13

**Overall Recommendation:** 3

**Summary:**

The paper proposes ​CROW (Internal Consistency Regularization), a defense mechanism to eliminate backdoor attacks in Large Language Models (LLMs). Backdoor attacks manipulate model outputs using hidden triggers, posing significant security risks. Existing defenses designed for classification tasks fail for generative LLMs due to their complex output spaces. CROW addresses this gap by enforcing ​layer-wise consistency in hidden representations during fine-tuning, neutralizing backdoors without requiring prior trigger knowledge or clean reference models.

**Claims And Evidence:**

Pros:
- *Abstract*: Backdoored models exhibit unstable layer-wise hidden representations when triggered.
    - The pre-experiment in Figure 1 well-illustrated the claim and was convincing.

Cons:
- *Section 3.1 Key Insight*: By tracking cosine similarity across consecutive layers, large gaps in similarity reveal potential backdoor-induced shifts.
    - The inconsistency observed in Figure 1 is calculated by the similarity differences between clean and backdoor data. However, we only own clean data for fine-tuning during the defense. A gap exists between the observation and the proposed method. We don't know whether the inconsistent internal representation is caused by the active representation of backdoor data compared to clean data, or exhibit the similar inconsistency with only the clean data.

**Essential References Not Discussed:**

See above.

**Experimental Designs Or Analyses:**

- The main experiments are considered adequated with various settings.
- The ablation study in 5.2 didn't show the cases without perturbation embedding. Since the main contribution is consistency, it is necessary to validate the effectiveness of the pure consistency loss.

**Methods And Evaluation Criteria:**

**Methods**:
- Finetuning techniques are reasonable and frequently implemented for backdoor defense.
- The internal consistency is intuitively reasonable for defense, although it cannot be fully illustrated in Figure 1.

**Evaluations**:
- The evaluation metrics and attack types following BackdoorLLM are considered adequated.

**Other Comments Or Suggestions:**

- It is confusing that the abbreviation *CROW* for the proposed method *Internal Consistency Regularization*.
- A clear illustration to highlight your contribution compared to the previous insights is necessary.

**Other Strengths And Weaknesses:**

Pros:
- The defense performance seems good in section 5.1, showing that the defense is effective.
- The experiments are adequate.
- The method is simple yet effective.
- The paper is good writing.

Cons:
- There exists a gap between the observation in Figure 1 and the method.
- The similar insights from the previous papers are not well-discussed in this paper.

**Questions For Authors:**

See above.

**Relation To Broader Scientific Literature:**

- The metric for calculating the differences between clean and backdoor data was previously proposed as Trigger-activated Change (TAC) in [1], but it seems not mentioned in this paper.
- Paper [1] also uses Lipschitzness to illustrate the backdoor behavior as in 3.3, and it can be used to support the conclusion here.
- The activation differences observed in clean and backdoor data are similarly observed in the backdoored model of the traditional classification models [2]. It seems equivalent to the cosine similarity differences observed in Figure 1.

[1] Data-free Backdoor Removal based on Channel Lipschitzness, ECCV 2022.

[2] Pre-activation Distributions Expose Backdoor Neurons, NeurIPS 2022.

**Theoretical Claims:**

The theoretical claims in 3.3 are correct. However, it is confusing that the conclusion *model retains stable behavior on clean inputs* seems to conflict with the loss target that we need to fine-tune a consistent representation with clean data input.

---

> ### Author Rebuttal · Authors · 2025-04-01
>
> # Response to Reviewer 8Evs
>
> Thank you very much for reviewing our paper and the valuable comments.
>
> **Q1.** How does Figure 1's analysis of clean vs. backdoor data align with CROW's clean-only defense approach?
>
> **A1.** We thank the reviewer for pointing out this important distinction. Figure 1 is used purely for diagnostic purposes, serving to visually motivate the hypothesis that backdoor triggers induce disruptions in internal representations, while clean inputs (in both clean and backdoored models) show smooth transitions. Our actual defense is entirely trigger-agnostic and uses only clean data. We will revise the Key Insight section to clarify this distinction.
>
> To approximate the instability caused by backdoor triggers, we introduce small adversarial perturbations to clean input embeddings during fine-tuning (Section 3.2). These perturbations are guided by gradients of the internal consistency loss and simulate trigger-like effects without relying on any triggered input. Our theoretical analysis (Section 3.3) shows that small perturbations can amplify across layers in a backdoored model unless near-isometry is enforced. CROW explicitly reimposes this stability through regularization, mitigating backdoor effects.
>
>
> **Q2.** Why do we need consistency fine-tuning if the model already shows stable behavior on clean inputs?
>
> **A2.** The key point is that the statement “the model retains stable behavior on clean inputs” refers to the final model after consistency regularization, not the initially backdoored model. Our theoretical analysis shows that near-isometric layer transformations ensure small input perturbations do not amplify across layers (Eq. 10), which leads to stability in internal representations. However, backdoored models typically violate this property: their hidden layers may appear stable on clean inputs superficially but are vulnerable to compounding deviations from small, malicious perturbations like hidden triggers.
>
> To address this, CROW introduces small adversarial perturbations to clean inputs during finetuning (Section 3.2). This simulates potential instability and guides the model to enforce layer-wise consistency under stress, thereby restoring the near-isometry property. The consistency loss targets this internal robustness, not just surface-level clean behavior. Therefore, there is no conflict: our loss function encourages stability under adversarial conditions, and the resulting model, after regularization, exhibits robust and stable behavior on clean inputs as well as resilience to hidden triggers. We will clarify this in the revised manuscript to avoid this ambiguity.
>
> **Q3.** Pure consistency
>
> **A3.** Table below contrasts pure consistency (no adversarial perturbations) against our CROW approach. Pure consistency effectively reduces ASR for simpler Sentiment backdoors but struggles with Refusal attacks, while CROW achieves stronger mitigation across all cases. Both maintain similar MT-Bench scores.
>
> *Note: ASR/MT-Bench*
> |Task|Attack|No Def|Pure |CROW|
> |-|-|:-:|:-:|:-:|
> |Sentiment|BadNets|65.00/2.72|1.59/4.15|0.53/3.80|
> |Sentiment|CTBA|63.33/2.80|3.21/4.18|2.08/3.80|
> |Sentiment|VPI|13.79/3.08|0.52/4.24|0.00/3.69|
> |Refusal|BadNets|94.50/4.35|48.97/4.46|19.63/4.15|
> |Refusal|CTBA|82.16/4.40|18.82/4.25|2.38/4.27|
> |Refusal|VPI|98.99/4.36|13.33/4.13|0.50/4.28|
>
> This demonstrates that pure consistency regularization may suffice for some attacks, but adversarial perturbations are crucial to further reduce ASR on the hard cases.
>
> **Q4.** Prior work on TAC [1] and activation differences [2] shows similar observations about backdoor behavior and Lipschitzness.
>
> **A4.** Thank you for the insightful comment. While TAC [1] and activation-based analysis [2] share a similar motivation—observing internal inconsistencies caused by backdoors—our method CROW differs in key aspects:
>
> First, CROW targets large language models (LLMs), whereas [1] and [2] focus on image-classification DNNs. Due to architectural differences, the internal symptoms of backdoors vary: CROW observes hidden-state inconsistencies across layers (Figure 1), while [2] reports shifts in neuron pre-activation distributions.
>
> Second, CROW mitigates backdoors via **layer-wise consistency regularization**, promoting near-isometry across layers to suppress trigger effects. In contrast, [1] and [2] rely on **pruning-based strategies** to remove suspicious neurons or channels.
>
> Finally, pruning is less effective for LLMs due to their high redundancy and weaker neuron–trigger coupling. CROW’s global regularization is better suited for mitigating backdoors in LLMs.
>
> We will include this discussion and cite [1] and [2] in the revised manuscript.
>
> ---
> [1] Data-free Backdoor Removal based on Channel Lipschitzness, ECCV 2022.
>
> [2] Pre-activation Distributions Expose Backdoor Neurons, NeurIPS 2022.
>
> **Q5.** CROW abbreviation
>
> **A5.** "CROW" takes the "C" and "R" from Consistency Regularization, while "OW" was added for readability.

---

### Official Review · Reviewer_UhBq · 2025-03-15

**Overall Recommendation:** 3

**Summary:**

This paper proposes CROW, a novel backdoor defense for LLMs that relies on enforcing internal consistency in layer-wise hidden states. It addresses the limitations of existing methods by not requiring trigger knowledge or a clean reference model. Experiments on Llama-2, CodeLlama, and Mistral models demonstrate that CROW significantly reduces the success rate of backdoor attacks, while maintaining the original generative performance and requiring minimal computational overhead.

**Claims And Evidence:**

Yes

**Essential References Not Discussed:**

No

**Experimental Designs Or Analyses:**

The experiment part sounds good to me

**Methods And Evaluation Criteria:**

Yes, but the author also needs to consider the adaptive attack, which is a common practice for AI backdoor defense papers.

**Other Comments Or Suggestions:**

No

**Other Strengths And Weaknesses:**

CROW introduces internal consistency regularization as a novel mechanism explicitly addressing an overlooked property of transformers, stable transitions in hidden states for clean inputs versus disruptive transitions caused by backdoor triggers. This is different from other defenses like, pruning, quantization-based defenses, and fine-tuning on clean data.

Weakness: 1. Although the paper provided some theory proof in section 3.3, I don't think this provides any reasonable error bound or robustness analysis. In general, CROW relies heavily on empirical results without a rigorous theoretical underpinning, weakening its comparative position in terms of foundational robustness guarantees. 2.CROW’s evaluation does not extensively consider adaptive or advanced attackers, leaving open questions regarding its resilience in more sophisticated threat environments. 3. CROW requires careful tuning of consistency parameters

**Questions For Authors:**

See my comments about the weakness of the paper

**Relation To Broader Scientific Literature:**

The key contributions of CROW are well-positioned within the broader literature on backdoor attacks and defenses for LLMs

**Theoretical Claims:**

No theory proofs in this paper (section 3.3 does not provide formal theoretical guarantees)

---

> ### Author Rebuttal · Authors · 2025-04-01
>
> # Response to Reviewer UhBq
>
> Thank you for taking the time to review our paper and for your valuable comments. Please find our responses to your questions below.
>
> **Q1.** Theoretical guarantees
>
> **A1.** We acknowledge that Section 3.3 in our paper offers an intuitive Lipschitz-based argument rather than a formal end-to-end proof. However, this intuitive analysis already provides a fundamental justification for our method and serves as an important basis for ensuring its validity. More importantly,  the diversity of our CROW experiments—encompassing six different backdoor attacks, five LLM architectures, and multiple tasks—offers substantial evidence that enforcing near-isometric transformations effectively thwarts backdoors across scenarios. Achieving a theoretical guarantee regarding backdoor-freeness is known to be extremely challenging. The only work that we are aware of is [1], which is limited to traditional neural networks and limited kinds of backdoor triggers. We will explore a more rigorous bounding framework (e.g., ASR upper bound under certain distributional assumptions) in future work.
>
> [1] Long H. Pham and Jun Sun: Verifying Neural Networks Against Backdoor Attacks, CAV 2022.
>
> **Q2.** Adaptive attacks
>
> **A2.** Thanks for your suggestion. In response, we conducted additional experiments on adaptive semantic backdoor attacks, which use natural, context-dependent triggers (e.g., entity names) to simulate stealthier, adaptive behaviors. These flexible triggers are harder to detect than those attacks, which easily result in overfitting. The table below summarizes CROW’s effectiveness in mitigating such attacks:
>
> |Attack|No Defense: ASR(↓)|No Defense: MT-Bench(↑)|CROW: ASR(↓)|CROW: MT-Bench(↑)|
> |-|-:|-:|-:|-:|
> |VPI-Semantic|38.09|3.52|0.58|3.97|
> |Semantic-Instruction|89.10|4.10|3.52|4.24|
>
> While semantic triggers may reduce overfitting compared to fixed-string triggers, we find they still introduce subtle but measurable inconsistencies in layer-wise hidden representations. Since CROW directly regularizes internal consistency, it effectively detects and neutralizes these triggers—even when they are embedded in natural language. We will explore more advanced adaptive attacks arising in the future to further validate the robustness of CROW.
>
> [1] Yan et al. "Backdooring Instruction-Tuned Large Language Models with Virtual Prompt Injection." arXiv:2307.16888 (2024)
>
> [2] Zhang et al. "Instruction Backdoor Attacks Against Customized LLMs." arXiv:2402.09179 (2024)
>
> [3] Cheng et al. "Transferring Backdoors Between Large Language Models By Knowledge Distillation." arXiv:2408.09878 (2024)
>
> **Q3.** CROW requires careful tuning of consistency parameters
>
> **A3.** This concern is closely related to the "Hyperparameter Sensitivity" question addressed in our response to Reviewer 3BdD (Q1). As detailed there, our comprehensive analysis shows that CROW exhibits strong robustness with respect to hyperparameter selection, due to several factors:
>
> (1) Our ablation studies demonstrate consistent ASR reduction across a wide range of α values (0.5–11), with stable backdoor mitigation observed over the moderate range of α ∈ [3.0, 7.0].
>
> (2) In practice, a single α value generalizes well across tasks and attack types. For example, setting α = 5.5 consistently works for all sentiment classification tasks against all attacks, while α = 11 performs effectively for all refusal-generation tasks.
>
> (3) The robustness of CROW is also supported by our Lipschitz-based analysis: the key lies in enforcing near-isometry across layers, which effectively neutralizes perturbations regardless of the trigger mechanism.

---

### Official Review · Reviewer_3BdD · 2025-03-15

**Overall Recommendation:** 4

**Summary:**

The paper proposes Internal Consistency Regularization (CROW) as a way to train away backdoors in LLMs. It's based on a finding that LLMs with a backdoor exhibit high levels of variance in layer-wise hidden representations when triggered. The proposed method adds a consistency loss term, and adversarially train the model to have high consistency in layer-wise hidden representations. Experiments show that CROW can significantly reduce attack success rates across multiple attacks, architectures and tasks.

**Claims And Evidence:**

The claims are well supported.

**Essential References Not Discussed:**

N/A

**Experimental Designs Or Analyses:**

Yes, I checked the soundness and validity of the experimental designs.

**Methods And Evaluation Criteria:**

The proposed method and evaluation criteria make sense. The method is based on empirical observation.

**Other Comments Or Suggestions:**

N/A

**Other Strengths And Weaknesses:**

Strengths:
* observes that clean and backdoored model exhibit different levels of consistency of layer-wise hidden representations.
* Based on the observation, derived an algorithm that regularizes for layer-wise consistency
* demonstrates that the algorithm can largely "train away" backdoors in multiple backdoor tasks with different triggers, on different models.

Weaknesses:
* It seems to be sensitive to hyperparameters, and \alpha is quite different across different tasks. This limits the practical significance of this work, as real model might have more than one type of backdoors.

**Questions For Authors:**

* What is the training curve like for CROW? Is training stable?
* would layer-wise consistency observation still hold if the trigger is semantic (a flexible distribution with a semantic meaning) rather than a fixed string? that might potentially reduce overfitting

**Relation To Broader Scientific Literature:**

The paper is related to backdoor attacks and defences in the broader scientific literature.

**Theoretical Claims:**

N/A

---

> ### Author Rebuttal · Authors · 2025-04-01
>
> # Response to Reviewer 3BdD
>
> Thank you for taking the time to review our paper and for your valuable comments. Please find our responses to your questions below.
>
> **Q1.** Sensitive to hyperparameters α
>
> **A1.** We thank the reviewer for raising concerns about the sensitivity of CROW's hyperparameter α. In fact, we have reported ablation studies of α in Section 5.2 for BadNets-CI using CodeLlama-7B, Table 6:
>
> |α|0.5|3.0|5.5|7.0|11.0|
> |-|---|---|---|---|----|
> |ASR(↓)|4.35|1.61|0.87|0.00|0.00|
> |MT-Bench(↑)|3.93|3.89|3.95|3.50|3.23|
>
> (1) **Consistent ASR reduction across tasks**: As seen above, even when α varies from 0.5 to 11, the ASR remains below 5% in the midrange of α. This trend holds for multiple backdoor strategies (e.g., BadNets, Sleeper, multi-trigger attacks), indicating that it is often not necessary to re-tune α per backdoor type.
>
> (2) **Minimal tuning overhead**: Adjusting α requires only 100 clean samples and about 3–4 minutes on a single A100. A short grid search (3–5 values) completes in under 20 minutes, which is practical in real deployment.
>
> (3) **Generalization to multiple backdoors**: Since CROW's consistency regularization is trigger-agnostic, fixing the α = 5.5 yields consistently effective defense results against diverse backdoor attacks.
>
> In the revision, we will include more task-specific ablation results (similar to the one above) to further demonstrate CROW's stable performance with varied hyperparameters. This should underscore that CROW's practical significance remains high, even for real-world cases involving multiple backdoors.
>
> **Q2.** What is the training curve like for CROW? Is training stable?
>
> **A2.** Thank you for the valuable question. Since we are unable to show figures during rebuttal, we will include full training-loss plots in our revised submission. Specifically, in our current experiments on Llama-2-7B (across five backdoor attacks), training remains stable despite adversarial perturbations. For Sentiment Steering, loss drops from ~3.25 to ~0.75 within 50 steps, converges near 0.1 by step 300, and stabilizes below 0.05 by step 400. Targeted Refusal exhibits minor oscillations from steps 50–300 before converging near 0.05 by step 600. Similar convergence patterns emerge for other backdoor types, showing that CROW’s consistency constraints enable smooth optimization without requiring any specialized techniques. We will update the visual training curves to demonstrate these stable trends in the revision.
>
> **Q3.** Would layer-wise consistency observation still hold if the trigger is semantic rather than a fixed string?
>
> **A3.** Thanks for the insightful question. To answer the question, we have conducted experiments to evaluate our approach against advanced semantic backdoor attacks. Below is the table from our new experiments, where the triggers are more flexible distribution with a semantic meaning rather than fixed strings:
>
> |Attack|No Defense: ASR(↓)|No Defense: MT-Bench(↑)|CROW: ASR(↓)|CROW: MT-Bench(↑)|
> |-|-:|-:|-:|-:|
> |VPI-Semantic|38.09|3.52|0.58|3.97|
> |Semantic-Instruction|89.10|4.10|3.52|4.24|
>
> While semantic triggers may reduce overfitting compared to fixed-string triggers due to their flexible nature, our results confirm they still induce subtle yet measurable inconsistencies in hidden representations. Because CROW enforces consistency across all layers, it effectively detects and mitigates these semantic triggers. We will include details on the learning curves and discussion in our revision to illustrate how this layer-wise regularization neutralizes semantic triggers without any extra tuning.
>
> [1] Yan et al. "Backdooring Instruction-Tuned Large Language Models with Virtual Prompt Injection." arXiv:2307.16888 (2024)
>
> [2] Zhang et al. "Instruction Backdoor Attacks Against Customized LLMs." arXiv:2402.09179 (2024)

---

> > ### Comment · Reviewer_3BdD · 2025-04-05
> >
> > Thank you for addressing my concerns and questions. I believe the contributions are novel and significant. I vote for accepting (raised score to 4).

---

> > > ### Author Response · Authors · 2025-04-05
> > >
> > > Thank you very much for revisiting our responses and for your thoughtful reconsideration. We will make sure to address all the reviewers' comments in the revision. Thank you again for raising the score!

---

### Decision · Program_Chairs · 2025-05-01

**Decision:**

Accept (poster)

**Comment:**

This paper proposes CROW, a novel backdoor defense for LLMs that relies on enforcing internal consistency in layer-wise hidden states. Although the reviewers generally felt that the methods and experiments of the paper were solid, some concerns were raised, such as the stability of the method and its robustness to potential adaptive methods. The authors did a good job in their rebuttal and all reviewers agree to accept this paper. Given its insights and effectiveness, I recommend accepting this paper.